# Differences in Plasma 25-Hydroxyvitamin D Levels at Diagnosis of Celiac Disease and Type 1 Diabetes

**DOI:** 10.3390/nu16050743

**Published:** 2024-03-05

**Authors:** Monica Marino, Tiziana Galeazzi, Rosaria Gesuita, Salima Ricci, Carlo Catassi, Valentino Cherubini, Elena Lionetti

**Affiliations:** 1Department of Women’s and Children’s Health, G. Salesi Hospital, 60123 Ancona, Italy; monicamarino96@gmail.com (M.M.); valentino.cherubini@gmail.com (V.C.); 2Department of Pediatrics, Marche Polytechnic University, 60100 Ancona, Italy; t.galeazzi@gmail.com; 3Center of Epidemiology and Biostatistics, Polytechnic University of Marche, 60123 Ancona, Italy; r.gesuita@staff.univpm.it; 4Department of Pediatrics, Women’s and Children’s Health, Azienda Ospedaliero-Universitaria Ospedali Riuniti Ancona, Marche Polytechnic University, 60121 Ancona, Italy; salima.ricci@gmail.com (S.R.); c.catassi@univpm.it (C.C.)

**Keywords:** celiac disease, type 1 diabetes, vitamin D status

## Abstract

Aim: The aim of this work is to assess the vitamin D levels, evaluated as plasma 25-hydroxyvitamin D of children with a new diagnosis of celiac disease (CD), of children with a new onset of type 1 diabetes (T1D) and in children with CD at diagnosis of T1D (T1D&CD). Methods: In this single-center observational study, we collected data for four groups of children and adolescents: T1D, CD, T1D&CD, and a control group (CG). The CG included schoolchildren who had negative results during a mass screening campaign for CD and were not diagnosed for T1D, according to RIDI Marche registry data, were considered for the purposes of this study. Plasma 25-hydroxyvitamin D, 25(OH)D_2_, and 25(OH)D_3_ were considered as the parameters for evaluating vitamin D nutritional status, and the date of measurement was recorded to analyze vitamin D level seasonality. Vitamin D nutritional status was categorized as follows: severe deficiency (<10 ng/mL), deficiency (<20 ng/mL), insufficiency (20–29 ng/mL), or sufficiency/adequacy (≥30 ng/mL). The Kruskal–Wallis test was used to compare the groups. The association of 25(OH)D levels with health conditions and seasonal differences of 25(OH)D levels was analyzed using a multiple linear regression model. Results: The number of children enrolled for the present study was 393: 131 in the CG, 131 CD, 109 T1D, and 22 T1D&CD. Significantly lower levels of vitamin D were displayed for children with CD, T1D, or both the diseases. Interestingly, severe vitamin D deficiency was detected in no children with CD, 1.5% of children in the CG, in 24.4% with T1D, and 31.8% with T1D&CD (*p* < 0.001). As expected, the CG children vitamin D levels were significantly influenced by seasonality. Contrarily, no seasonal differences were reported in children with CD, T1D, and T1D&CD. Multiple regression analysis showed that children with T1D and T1D&CD had lower 25(OH)D levels of 9.9 ng/mL (95% CI: 5.4; 14.5) and 14.4 ng/mL (95% CI: 6.2–22.7) compared to CG children (*p* < 0.001). Conclusions: Our results showed low levels of vitamin D diagnosis of T1D, CD, and T1D&CD; however, severe deficiency was only reported in children with T1D and T1D&CD. More studies are needed to better understand the role of this deficiency in children newly diagnosed with CD and T1D.

## 1. Introduction

Celiac disease (CD) is a systemic, permanent, autoimmune-mediated disease caused by the reaction to proline- and glutamine-rich proteins contained in rye, barley, wheat, in genetically susceptible individuals [1]. CD is significantly associated with other autoimmune diseases, particularly type 1 diabetes (T1D). 

While CD affects approximately 1% of the population worldwide [2], it varies between 1.6% and 9.7% in patients with T1D [3]. The latter disease is an endocrine metabolic disorder and a consequence of the autoimmune-mediated destruction of the pancreatic beta cells, manifesting with insulin deficiency [4].

In the last few decades, lifestyle and environmental changes most likely played a role in increasing the prevalence of CD and T1D in western countries [5]. Diet, infections, toxins, and humoral status have been recognized as candidate factors that affect children in early stages of life in utero, perinatally, or during early childhood. Above all, although several factors have been investigated, due the multifactorial nature of the disease, these factors might be considered not singularly but as being the result of the coexistence of more factors that, if combined, strongly increase the risk of developing the disease. 

Vitamin D plays an important role in regulating the autoimmune system, and several studies have previously considered it to be a protective factor against the development of autoimmune disease. An association between vitamin D deficiency and severity of different autoimmune diseases has been reported in the literature, among them lupus disease, inflammatory bowel disease, type 2 diabetes, multiple sclerosis, CD, and T1D [6,7,8,9,10,11]. In addition, patients with a lower 25(OH)D3 level are more likely to be exposed to autoimmune disease onset [6,7,8,9,10,11]. It has been reported that children with CD have significantly lower levels of vitamin D at disease diagnosis then the general population [12]. Vitamin D deficiency has also been reported in T1D childhood onset [13].

For instance, the role of vitamin D gene receptor polymorphism has been investigated as a risk factor for vitamin D deficiency, as it is correlated with the low production of physiological endogenous vitamin D [12]. Polymorphism in genes encoding the vitamin D receptor has been previously documented in children with CD [14], as well as in children with T1D [15].

A further hypothesis was the association between increasing prevalence in autoimmune disease, including T1D and CD, and increasing latitude, with the decrease in sunlight exposure, and, consequentially, in vitamin D production [16,17]. 

Furthermore, the association between the risk of developing both CD and T1D and month of birth corroborates the link between sun exposure and autoimmunity [16,18,19,20,21,22,23]. As regards CD and T1D incidence in the northern hemisphere, a higher risk was reported for children born in April, while a lower risk was observed for children born between October and November.

Indeed, the supplementation of vitamin D during early life stages could be correlated with a reduced risk of T1D onset [24]. Nevertheless, results from a recent meta-analysis demonstrated that this association should be verified to be considered more than a hypothesis and highlighted the lack of evidence to prove an actual association between maternal vitamin D supplementation and T1D incidence [25]. 

There is a lack of studies considering vitamin D levels in children with CD and T1D. A recent cross-sectional study from the United States, that recruited 62 obese children, evaluated vitamin D status, highlighting that the coexistence of both T1D and CD in this population may be associated with lower 25-OHD in comparison with healthier controls [26]. 

The main aim of this study is to evaluate and compare vitamin D nutritional status in children with a new diagnosis of CD, in children with a new onset of T1D, and in children with CD that are newly diagnosed with T1D. 

## 2. Materials and Methods

### 2.1. Study Design and Population

The protocol of the present study was approved by the Independent Ethical Committee of Marche Region on 18 June 2020 (protocol number 2020 49, OR study ID 124827). This observational study was conducted by the Center for Gastroenterology and the Center for Pediatric Diabetology of the Polytechnic University of Marche from October 2020 to March 2022. Ancona is the regional capital city of the Marche, a region of 1,525,000 inhabitants (year 2019) in a geographical area of 9694 square kilometers. It is in the center of Italy with an annual sunlight exposure of 2220 h. November is the darkest month, while July is the lightest. 

Children and adolescents included in the present study were asked to sign an informed consent form together with their parents or caregivers. We enrolled participants aged 2–18 years divided into 4 groups: (1) CD group, (2) T1D group, (3) T1D&CD group, and (4) control group (CG). 

Data on the T1D group and T1D&CD group were population-based and collected from the RIDI Marche Registry [27,28]. RIDI was set up in 1997 to coordinate registries for the incidence of type 1 diabetes in Italy. All registries anonymously report clinical information on newly diagnosed insulin-treated children from 1997. All patients with T1D and complete available data in the RIDI Marche registry since October 2018 were included in these analyses. The new diagnosis of T1D was based on the international guidelines criteria [29]. 

We compared two groups with the control group (CG) of children not affected with CD (according to the criteria of the European Society for Paediatric Gastroenterology Hepatology and Nutrition [30]) or T1D according to the RIDI Marche registry and a group of children with CD (CD group) that have been previously described [12]. These data were collected during a mass screening program for CD that involved 5705 school-aged children. Blood test results from this campaign were analyzed for a previous case–control study, conducted from October 2018 to January 2020, where cases were children diagnosed with CD (based on a positive result of the IgA class anti-transglutaminase test) and controls were children matched for clinical characteristics who reported negative results in CD antibody screening [12]. Plasma 25-hydroxyvitamin D (25-OHD_2_ and 25-OHD_3_) was also measured from blood samples collected.

Exclusion criteria were (a) associated chronic conditions (Hashimoto thyroiditis, inflammatory bowel disease) or genetic disorders (Down or Turner syndrome); (b) use of vitamin D supplements during the last 6 months.

Vitamin D status data and other clinical variables were collected for the purpose of the present study for all study participants. 

### 2.2. Vitamin D Assessment

The plasma concentration of 25-hydroxy vitamin D (25-OHD), including 25-hydroxy vitamin D_2_ and 25-hydroxy vitamin D_3_, was selected as biomarker of vitamin D nutritional status, and it was quantitatively determined by a chemiluminescent immunoassay (LIAISON, DiaSorin, Saluggia, Italy). Annual vitamin D status assessment has been recognized by The Italian Society of Pediatric (SIP) guidelines as a routine test for children at risk of developing vitamin D deficiency [31]. Vitamin D deficiency status was defined as follows: severe deficiency (<10 ng/mL), deficiency (<20 ng/mL), insufficiency (20–29 ng/mL), or sufficiency/adequacy (≥30 ng/mL) [31].

For children with CD and for children with T1D only, the 25(OH)D concentration was assessed at diabetes diagnosis. For children with T1D&CD, 25(OH)D levels were assessed at the diagnosis of T1D, in the presence of CD. The month of blood samples collection was recorded and classified as 1 of 4 seasons: winter, spring, summer, or autumn. The Winter season considered includes all days between December and February (sunlight exposure 9.14 h/day–11.06 h/day); the spring season includes all days between March and May (sunlight exposure 11.12 h/day–15.11 h/day); the summer season includes all days from June to August (sunlight exposure 13.19 h/day–15.12 h/day); the autumn season includes all days from September to November (sunlight exposure 9.14 h/day–13.15 h/day). 

### 2.3. Other Clinical Variables 

Gender, age, and anthropometric measurements were collected for the purposes of the present study. Weight and height were measured using the same mechanical scale (SECA 200) and stadiometer (SECA 220) for all children included. Participants were divided into one of the following three groups based on BMISDS: underweight (BMI-SDS < 0), normal weight (BMI-SDS 0 to < 1.28), or overweight (BMI-SDS > 1.28), according to WHO growth curves [32]. 

### 2.4. Statistical Analysis

The characteristics of the children in each group of health conditions were summarized based on the distribution of the variables. Qualitative variables were expressed as absolute frequency and percentage, and comparisons between groups were assessed using Fisher’s exact test. Quantitative variables were summarized using the median as a measure of centrality and the interval between the first and third quartiles (IQR) as a measure of variability; comparisons between groups were evaluated using the Kruskal–Wallis test. The multiple linear regression model was used to analyze the association of the vitamin D concentration (dependent variable) with health conditions, the season to vitamin D measurement, and their interaction (independent variables), adjusted by children’s BMI-SDS. The multiple linear regression model assumes a linear association between the dependent and the independent variables and allows the estimation of regression coefficients that measure the expected change in the dependent variable due to a unit change in the independent variable. It also allows the estimation of interaction terms, i.e., the synergistic effect of the season of measurement and health condition. The CG was the reference category. The estimated regression coefficients, 95% confidence intervals (95% CI) and *p*-values for each independent variable were reported. R^2^ was used as a measure of the goodness of fit of the model to the data.

All analyses were performed using the R version 4.04 statistical software. 

## 3. Results

Demographic and clinical characteristics of the four groups of study are shown in Table 1. Overall, 393 children were included in the analysis. 

Figure 1 shows the distribution of 25(OH)D in the four study groups. Subjects in the CG had a median value of 29 ng/mL (IQR 23–35), in CD group the median value was 24 ng/mL, in T1D the median was 16 ng/mL (IQR 10–18), while in T1D&CD group the median value was 14 ng/mL (IQR 9–20). The 25(OH)D levels in children with T1D or CD or with T1D&CD were significantly lower than the CG (Figure 1). A significant difference was also found between children with T1D or with T1D&CD as compared with children with CD. 

In TD1 and T1D&CD groups, there was a significantly higher percentage of severe deficiency and deficiency of vitamin D as compared with the CG and CD groups. In the CG, there was a significantly higher percentage of children with sufficiency as compared with the CD, T1D, and T1D&CD groups (Table 1). 

The distribution of 25(OH)D levels in each group of children was analyzed according to the season of blood extraction (Figure 2). Vitamin D values were significantly higher in summer and in autumn than in winter and spring in children in the CG, as expected. Interestingly, the observed distribution of 25(OH)D in children with CD and/or T1D was stable and had very low variability between different seasons. 

Table 2 shows the results of multiple regression analysis, in which the interaction term between study groups and season was analyzed. Children with T1D and children with T1D&CD on average had lower vitamin D than CG children, of 10.0 and 14.2 ng/mL respectively. As expected, the vitamin D level of 14.6 and 16.6 ng/mL in summer and in autumn was significantly higher than in winter. The interaction terms showed that children with CD or T1D had significantly lower mean levels of 25(OH)D in summer (of 14.2 and 11.2 ng/mL, respectively) and autumn (of 15.7 and 9.8 ng/mL, respectively) as compared with the CG in the same seasons. No significant effect of BMI on vitamin D level was observed.

## 4. Discussion

To the best of our knowledge, this is the first study comparing vitamin D levels at diagnosis of CD, T1D, and both diseases. Although low 25(OH)D levels have been found for all these conditions, severe deficiency has only been reported in children with T1D and T1D and CD. Furthermore, while plasma levels of vitamin D vary with the seasons in children without CD and T1D, no seasonal variation was found in children with CD, T1D and both diseases. 

In the present study, children newly diagnosed with T1D or CD and children with both T1D and CD had significantly lower vitamin D levels when compared with control group children. This finding reinforces what was reported in a recent American study that evaluated the status of vitamin D in a population of obese children and adolescents in the presence of T1D, CD, and both diseases [26]. However, these analyses were performed on a different study population with obesity, which is reported to be a condition independently correlated with vitamin D deficiency, and vitamin D status was evaluated 6 months after disease diagnosis in a small sample of children. 

In addition, in our study, we categorized vitamin D status into four levels: sufficiency, insufficiency, deficiency, and severe deficiency. This allowed us to reveal a higher frequency of deficiency and severe deficiency in the T1D group and T1D&CD group vitamin D status when compared to both CD and the CG (62.4% and 77.3% vs. 3.8% and 13.7%, respectively).

Children with T1D display a lower level of serum 25(OH)D when compared with people without diabetes. 

A recent meta-analysis that included 24 observational studies, therefore 1900 children and 3494 adults with T1D in total, reported statistically significant median values of serum 25OHD levels: 2.61 ng/mL [1.13–4.09] for adults and 5.69 ng/mL [2.82–8.55] for children, highlighting a severe deficiency in vitamin D status for this study population. In addition, these findings are supported by other observational studies on worldwide population [12,13]. Among these eleven studies on children with T1D, only one excluded children with T1D and CD from the analyses, while in the other ten there were no specifics on the presence or absence of CD in the study populations [33]. 

For children with CD, among the published studies, several appeared as heterogeneous in their design and outcomes. Our study results are in line with what was already reported in the literature as regards low vitamin D levels at diagnosis [34,35]. 

In the present study, blood sampling dates were recorded to assess the variability of endogenous vitamin D production throughout the year. The results of our study revealed that the distribution of vitamin D status among the seasons is variable in the CG, according to sun exposure duration, as vitamin D values were significantly higher in summer and in autumn than in winter and spring. However, no significant differences in vitamin D levels among seasons were observed in the presence of the autoimmune diseases considered. 

Previous studies considered vitamin D deficiency as a consequence of malabsorption in children and adolescents with CD. The correlation between CD and the osteoporosis and fracture risk has been investigated previously [36,37]. However, the involvement of early-life vitamin D deficiency in primary stages of the pathogenesis of CD should be reconsidered regarding the role of vitamin D in immune system regulation [38]. 

Specifically, vitamin D is involved in the regulation of the immune system by influencing the inhibition of the differentiation and maturation of dendritic cells [39,40]. Antigen presentation to a T cell by a mature dendritic cell facilitates an immune response, whereas antigen presentation by an immature dendritic cell facilitates tolerance. Vitamin D deficiency is associated with a reduced inhibition of dendritic cell maturation, with consequential increasing T-cell responses. In the context of autoimmunity, this mechanism is very important as self-antigens are abundant in physiologic state, and presentation of these self-antigens may occur abnormally by matured dendritic cells [39]. Many tissues express the vitamin D receptor, including immune cells [41]. The 1α-hydroxylase (CYP27B1) gene regulates the expression of the vitamin D receptor on immune system cells (B cells, T cells, and antigen-presenting cells) [14,42,43,44]. All these cells can respond not only to the active vitamin D metabolite but also to its precursors, and they are able to convert inactive vitamin D, 25-hydroxyvitamin D, into its active form, 1,25(OH)2D [45].

Vitamin D gene receptor polymorphism has been investigated as a risk factor for vitamin D deficiency, as it is correlated with the low production of physiological endogenous vitamin D [12,46,47,48]. Polymorphism in genes encoding the VRD or enzymes involved in vitamin D synthesis (DHCR7, CYP2R1, and CYP27B1) has been documented in children with CD [39]. Therefore, the very low variability observed within each season of vitamin D in children with CD may be explained by a reduced endogenous production of 1,25(OH)2D due to polymorphism in genes encoding the vitamin D receptor in immune system cells [14,15,49]. The vitamin D receptor gene has also been investigated in T1D, and a positive association between haplotype “fBAt” of the vitamin D receptor gene and risk for T1D has been reported [15]. 

Even though T1D and T1D&CD groups displayed no statistically significant variability in vitamin D status throughout seasons in our results, the role of seasonality in sun exposure has been discussed in the literature for children and adolescents with T1D. 

The seasonality of birth in children with T1D could be explained by seasonal variation in endogenous production of vitamin D according to sun exposure [16]. Data collected from all Welsh pediatric units from 1990 to 2019 showed seasonal variability’s role in T1D onset and a preventative role for vitamin D both during pregnancy and later childhood [50]. Sunlight exposure seems likely to contribute to seasonal variability of T1D onset, from one to three cases per 100,000 children in China and some Asian and South American nations to 30–60 per 100,000 in Scandinavia [27,51]. A cross-sectional study from Belgium reported that average daily hours of sunshine were inversely related to the number of new patients with T1D [52]. However, a meta-analysis of observational studies of vitamin D intake during pregnancy showed no effect on the incidence of T1D [25].

Although vitamin D status in children appears to be a protective factor for the onset of T1D, studies of prenatal and childhood supplementation have produced controversial results, with difficult application in clinical practice.

The meta-analyses published by Zipitis et al. and Dong et al., as well as the one including observational studies from the EURODIAB project [24,25], suggested that vitamin D supplementation in childhood is associated with lower risk of developing T1D later in life, and it has been considered more effective than prenatal supplementation in this population The reduction in T1D risk estimated was 1.5-fold in both studies considered. 

Further results from randomized controlled trials differentiated the effect of supplementation of the active form of vitamin D, 25(OH)D3, from the effect of other forms of vitamin D supplementation, in terms of its benefits in treating children with a new diagnosis of T1D [53,54,55]. However, the supplementation with alfacalcidol (1-alpha-hydroxycholecalciferol), a vitamin D analogue, which is converted to calcitriol by hepatic metabolism, seems to preserve pancreatic function on in T1D onset in children and adolescents [56]. However, no association between vitamin D deficiency and T1D progression was detected in children with reported autoimmunity [57].

Children and young adults with newly diagnosed CD benefit from vitamin D and calcium supplementation to improve their bone mineral density in a short period of six months with a strict gluten-free diet [58,59]. There is a paucity of studies that investigate the association between vitamin D supplementation and prevention of CD.

Interestingly, the concept of the personal vitamin D response index has been recently presented, and vitamin D supplementation seems to depend on an individual’s personal vitamin D response index rather than on vitamin D status alone [60]. According to this concept, individuals can be categorized according to their vitamin D sensibility response into high, mid, and low responders by measuring vitamin D sensitivity parameters.

Further studies are required to explore the variety of options in vitamin D supplementation.

The main strength of the present study is the possibility to analyze the association between 25(OH)D and CD or T1D diagnosis in a sample of subjects with CD with newly diagnosed T1D. In addition, a population-based registry was used for groups with diabetes. However, the sample size of children with T1D&CD is limited. Further limitations are related to the cross-sectional nature of the study design that does not include information on plasma 25(OH)D levels before diagnosis of CD or T1D.

## 5. Conclusions

Our results showed low levels of vitamin D at diagnosis of T1D, CD, and T1D&CD; however, severe deficiency was only reported in children with T1D and T1D&CD. Furthermore, no seasonal variation was found in children with CD, T1D, or both diseases, suggesting a possible deficiency in endogenous vitamin D production or unavailability to convert endogenous vitamin D in its active form in this population.

This observational study allowed the comparison of two groups of participants (with T1D and with CD) in which vitamin D insufficiency has already been found at diagnosis. Furthermore, an additional group of children with CD who subsequently also had a diagnosis of T1D has been described, and significantly lower levels of 25-OHD are reported in this study population. Even though T1D and CD share an autoimmune pattern, severe deficiency of vitamin D levels was reported only in the presence of T1D.

Differences in seasonal variability of vitamin D levels for each group have been reported and described in this manuscript, highlighting a very low variability for the CD group of children and adolescents. Future research is required to confirm these differences between T1D, CD, and T1D&CD populations.

However, interventional and observational studies on vitamin D supplementation in T1D and CD populations have produced controversial evidence for both prevention and treatment of these diseases. Further studies are necessary to pave the way for adequate and personalized intervention strategies in the context of children and adolescents with T1D and CD, taking these differences and peculiarities into account.

## Figures and Tables

**Figure 1 nutrients-16-00743-f001:**
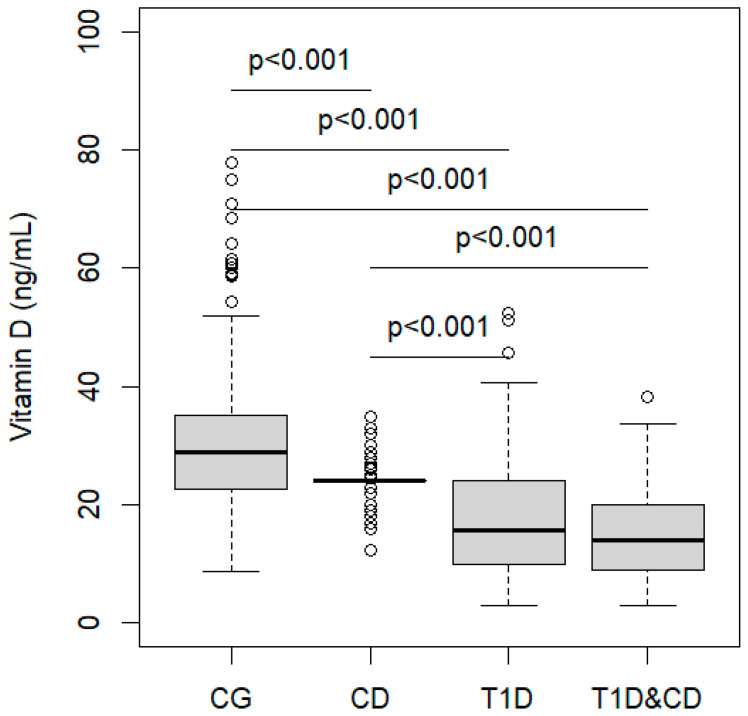
Vitamin D distribution in the four groups. Kruskal–Wallis test. *p*-values refer to multiple comparisons. CG: control group; CD: celiac disease; T1D: type 1 diabetes.

**Figure 2 nutrients-16-00743-f002:**
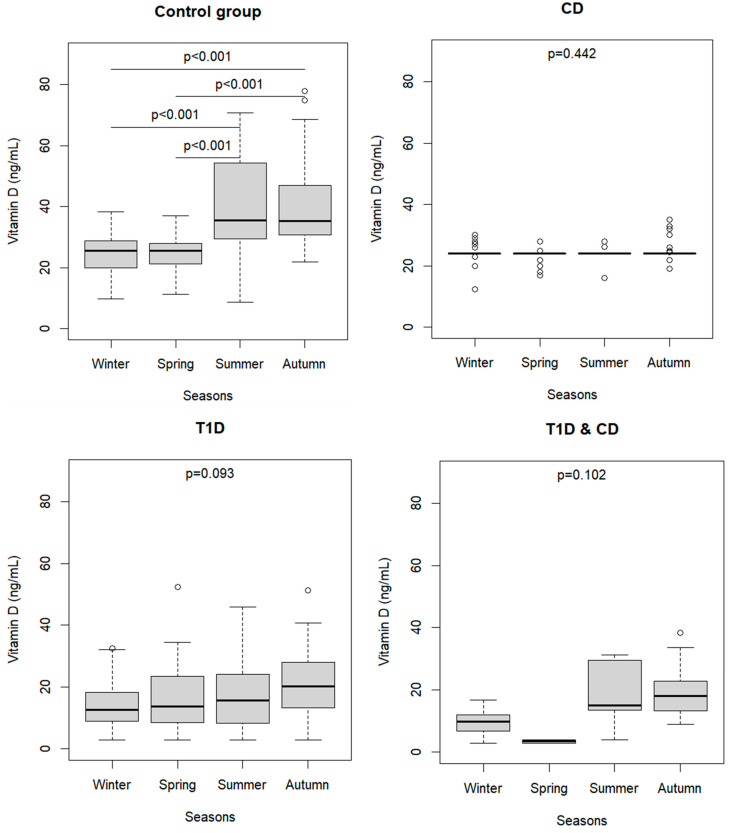
Vitamin D distribution in the four groups by seasons. Kruskal–Wallis test and multiple comparisons. Control Group; CD: celiac disease; T1D: type 1 diabetes.

**Table 1 nutrients-16-00743-t001:** Characteristics of subjects according to health conditions.

	CG (n = 131)	CD (n = 131)	T1D (n = 109)	T1D&CD (n = 22)	*p*
Gender, male, n (%)	50 (38.2)	50 (38.2)	64 (58.7)	12 (54.5)	0.003
Age (years), median (IQR)	8 (7–9)	8 (6–11)	10 (7–13)	8 (4–12)	<0.001 ^#^
Vitamin D classes, n (%)					
Severe deficiency	2 (1.5)	0 (0)	27 (24.8)	7 (31.8)	<0.001
Deficiency	16 (12.2)	5 (3.8)	41 (37.6)	10 (45.5)	
Insufficiency	58 (44.3)	121 (92.4)	29 (26.6)	2 (9.1)	
Sufficiency	55 (42)	5 (3.8)	12 (11)	3 (13.6)	
BMI in classes					
Underweight (BMI-SDS < 0)	100 (76.3)	33 (68.8)	72 (66.1)	15 (68.2)	0.135
Normal weight (0 ≤ BMI-SDS ≤ 1.28)	29 (22.1)	15 (31.2)	29 (26.6)	6 (27.3)	
Overweight (BMI-SDS > 1.28)	2 (1.5)	0 (0)	8 (7.3)	1 (4.5)	
BMI (Kg/m^2^), median (IQR)	17 (15–18)	17 (15–19)	17 (15–20)	6 (14–18)	0.305

Fisher’s exact test. ^#^ Kruskal–Wallis test, multiple comparisons: T1D vs. CG *p* < 0.001; T1D vs. CD *p* < 0.001); CG: control group; CD: celiac disease; T1D: type 1 diabetes.

**Table 2 nutrients-16-00743-t002:** Association of vitamin D with health conditions and seasons. Results from multiple linear regression analysis.

	Estimate	95% CI	*p*
Intercept	22.1	15.1; 29.1	<0.001
CD	−0.8	−6.9; 5.3	0.793
T1D	−10.0	−15.1; −4.9	<0.001
CD and T1D	−14.2	−23.5; −4.9	0.003
Spring	0.8	−3.8; 5.4	0.731
Summer	14.6	9.3; 19.8	<0.001
Autumn	16.6	12.1; 21.1	<0.001
BMI (Kg/m^2^)	0.1	−0.2; 0.5	0.541
CD in Spring	−0.1	−8.6; 8.4	0.984
T1D in Spring	2.1	−5; 9.2	0.565
T1D&CD in Spring	−7.3	−24.2; 9.7	0.398
CD in Summer	−14.2	−24; −4.4	0.005
T1D in Summer	−11.2	−18.9; −3.4	0.005
T1D&CD in Summer	−5.7	−19.1; 7.7	0.403
CD in Autumn	−15.7	−25.1; −6.2	0.001
T1D in Autumn	−9.8	−16.7; −3	0.005
T1D&CD in Autumn	−6.3	−17.8; 5.3	0.284

Multiple linear regression analysis. Estimate: the regression coefficient estimate, which measures the expected change in the dependent variable due to a unit change in the independent variable; 95% CI: 95% confidence interval of the regression coefficients; R^2^ = 0.43, showing a sufficient goodness of fit of the model to data. Winter is the reference category for the season at vitamin D measurement. CG: control group, the reference category; CD: celiac disease; T1D: type 1 diabetes.

## Data Availability

Data are contained within the article.

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
