# Peer review of "Differences in Plasma 25-Hydroxyvitamin D Levels at Diagnosis of Celiac Disease and Type 1 Diabetes"

_nutrients, 2024, doi:10.3390/nu16050743_

Round 1
Reviewer 1 Report
Comments and Suggestions for Authors
This study is observational research on the plasma levels of 25-hydroxyvitamin d in newly diagnosed children with coeliac disease (CD), type 1 diabetes (T1D), and both T1D and CD (T1D+CD). The findings of this study are value references for interventions aimed at improving health outcomes among children and adolescents affected by CD or TID. However, the analysis of this study is relatively simple and basic since it solely examines vitamin D plasma concentrations among a range of ages from 2-18 across various diseases while also exploring seasonal influences on such concentrations. To enhance its value further, it would be beneficial if the author delved into greater detail regarding specific age groups such as those aged between 2-6 years old, 7-10 years old, 11-14 years old and 15-18 years old. Additionally, investigating the impact of dietary habits on plasma vitamin D concentrations among different groups of affected children, including gender (male and female), would provide more valuable insights for nutritional interventions targeting the health of these individuals.
Here are some specific comments on the manuscript:
1. The title of this paper should clearly state 25-hydroxyvitamin D3, rather than the general expression 25-hydroxyvitamin d, and the 'd' in the original title should be capitalized as 'D'
2. The title of this paper is 'Differences in plasma 25-hydroxyvitamin d levels at diagnosis of celiac disease and type 1 diabetes,' but throughout the text, it focuses on examining the concentration of vitamin D in plasma. Please provide additional explanations in both the “Introduction” and “Materials and Methods” sections to clarify the relationships between vitamin D and 25-hydroxyvitamin D3, as well as to explain why measuring the concentration of vitamin D in plasma instead of 25-hydroxyvitamin D3.
3. Please provide specific descriptions for each season, including the months it spans from and to, the unique climate characteristics during that period in the local area, and what the lighting conditions are like.
4. The abbreviations for “25-hydroxyvitamin d“ in this article are inconsistent. For example, in lines 18, 109, 110, and 148 it is written as 25(OH)D3, while in lines 59, 161, 168,184, and 257 it is written as 25(OH)D3. Additionally, in lines 65, 102, and 270 it is referred to as 25-OHD.
5. Table 2 should be marked with 'To be continued' at the end of the first half if it is interrupted, and 'Continuation' should be indicated at the beginning of the second half. Additionally, both sections of the table must include top and bottom horizontal lines to ensure completeness.
6. In all figures, it should be written as “mL”.
7. The caption below Figure 1 should be written in full as follows: CG: Control Group. Additionally, all captions below Tables and Figures should have consistent capitalization, for example, CG: Control Group; CD: Celiac Disease; T1D: Type 1 Diabetes.
8. The publication years of references 7, 12, 21, and 22 should be highlighted in bold.
Author Response
1) This study is observational research on the plasma levels of 25-hydroxyvitamin d in newly diagnosed children with coeliac disease (CD), type 1 diabetes (T1D), and both T1D and CD (T1D+CD). The findings of this study are value references for interventions aimed at improving health outcomes among children and adolescents affected by CD or TID. However, the analysis of this study is relatively simple and basic since it solely examines vitamin D plasma concentrations among a range of ages from 2-18 across various diseases while also exploring seasonal influences on such concentrations. To enhance its value further, it would be beneficial if the author delved into greater detail regarding specific age groups such as those aged between 2-6 years old, 7-10 years old, 11-14 years old and 15-18 years old.
1) Thank you for reviewing the article and for your suggestions to improve it.
However, age-stratified analysis was not possible due to the small number of observations in the group of patients with both diseases (22 participants), as indicated in the discussion.
2) Additionally, investigating the impact of dietary habits on plasma vitamin D concentrations among different groups of affected children, including gender (male and female), would provide more valuable insights for nutritional interventions targeting the health of these individuals.
2)This is a very interesting point that unfortunately we cannot analyze in this study. This study is an observational study that aims to describe vitamin D levels in children and adolescents with T1D, CD and both diseases, to describe the differences and peculiarities of each group included.
We will analyze the eating habits of the participants in the different groups in a dedicated study.
3) The title of this paper should clearly state 25-hydroxyvitamin D3, rather than the general expression 25-hydroxyvitamin d, and the 'd' in the original title should be capitalized as 'D'.
3) Thank you, the title has been updated as suggested.
4) The title of this paper is 'Differences in plasma 25-hydroxyvitamin d levels at diagnosis of celiac disease and type 1 diabetes,' but throughout the text, it focuses on examining the concentration of vitamin D in plasma. Please provide additional explanations in both the “Introduction” and “Materials and Methods” sections to clarify the relationships between vitamin D and 25-hydroxyvitamin D3, as well as to explain why measuring the concentration of vitamin D in plasma instead of 25-hydroxyvitamin D3.
4) For the present study, we considered the assessment of 25-hydroxy vitamin D, that includes 25-hydroxy vitamin D2 plus 25-hydroxy vitamin D3 plasma concentration, as routinary exam for our clinical practice recommended by Italian Society of Pediatric (SIP) guidelines.
5) Please provide specific descriptions for each season, including the months it spans from and to, the unique climate characteristics during that period in the local area, and what the lighting conditions are like.
5) We have updated the manuscript as suggested (highlighted in yellow) and we thank you very much for this precious suggestion.
6) The abbreviations for “25-hydroxyvitamin d“ in this article are inconsistent. For example, in lines 18, 109, 110, and 148 it is written as 25(OH)D3, while in lines 59, 161, 168,184, and 257 it is written as 25(OH)D3. Additionally, in lines 65, 102, and 270 it is referred to as 25-OHD.
6) We have updated the manuscript as suggested and we thank you very much for this precious suggestion.
7) Table 2 should be marked with 'To be continued' at the end of the first half if it is interrupted, and 'Continuation' should be indicated at the beginning of the second half. Additionally, both sections of the table must include top and bottom horizontal lines to ensure completeness.
7) Thank you. We have just inserted Table 2 in a separate page to avoid interruption in the middle.
8) In all figures, it should be written as “mL”.
8) Thank you. We corrected “ml” with “mL” in all figures
9) The caption below Figure 1 should be written in full as follows: CG: Control Group. Additionally, all captions below Tables and Figures should have consistent capitalization, for example, CG: Control Group; CD: Celiac Disease; T1D: Type 1 Diabetes.
9) Thank you very much for this suggestion. We have updated data according to you comment.
10) The publication years of references 7, 12, 21, and 22 should be highlighted in bold.
10) Thank you, we have updated data according to your suggestion.
Reviewer 2 Report
Comments and Suggestions for Authors
In the present study the authors assessed the Vit D levels in the newly diagnosed children with CD and TID and both CD & TID. The authors also assessed the effect of seasons on the Vit D level. However, the study has a lot of flaws and needs to be rectified before it can be recommended for publication. Below are my comments:
In the abstract, the authors mentions that severe Vit D deficiency was found in 1.5% of the CG, in 0% of children with CD, 24.4% with T1D, and 31.8% with T1D&CD 24 (p <0.001). So, are the authors concluding that control group had severe Vit D deficiency as compared to CD which had 0%? Please explain
Introduction needs to be rewritten. Explaining the role of VDR, calcitriol and their mechanism before mentioning them in the discussion section for better clarity.
25-Hydroxyvitamin D is written as 25(OH)2D3, 25(OH)D3, 25OHD, 25-OHD and sometimes 25(OH)D3. Needs consistency.
Author Response
1) In the abstract, the authors mentions that severe Vit D deficiency was found in 1.5% of the CG, in 0% of children with CD, 24.4% with T1D, and 31.8% with T1D&CD 24 (p <0.001). So, are the authors concluding that control group had severe Vit D deficiency as compared to CD which had 0%? Please explain
1) Thank you for revising the article and for the suggestions provided. Vitamin D deficiency status was defined as follows: severe deficiency (<10 ng/ml), deficiency (<20 ng/ml), insufficiency (20-29 ng/ml) and sufficiency/adequacy (≥30 ng/ml). Therefor the datum mentioned is true for this analysis.
We thank you very much for this precious comment, so that we add the opportunity to clarify in the abstract.
2) Introduction needs to be rewritten. Explaining the role of VDR, calcitriol and their mechanism before mentioning them in the discussion section for better clarity.
2) Thank you very much for your comment. We have added some insights of this topic in the Background, as suggested.
3) 25-Hydroxyvitamin D is written as 25(OH)2D3, 25(OH)D3, 25OHD, 25-OHD and sometimes 25(OH)D3. Needs consistency.
3) Thank you very much for your suggestion. For the present study, we considered the assessment of 25-hydroxy vitamin D, that includes 25-hydroxy vitamin D2 plus 25-hydroxy vitamin D3 plasma concentration, as routinary exam for our clinical practice recommended by Italian Society of Pediatric (SIP) guidelines. We have updated the data in the manuscript, as requested.
Round 2
Reviewer 1 Report
Comments and Suggestions for Authors
The authors have revised some of the comments suggested by referees, but there are still some points that need to be revised.
1. The correct abbreviation should be 'mL' instead of 'ml', and the authors should make this correction. For example, lines 20-21; line 30; lines 123-124; lines 159-161; lines 181; line 213.
2. The abbreviations for “25-hydroxyvitamin D“ from lines 73 to 81 are still inconsistent, and “3” for "D3" should be formatted as a subscript.
3. Line 221, it should be “[33, 34]”not “[33], [34]”.
4. All tables and figures should have consistent capitalization. CG: Control Group; CD: Celiac Disease; T1D: Type 1 Diabetes.
5. The publication years of references 11, 13, 22 and 23 still need to be revised, which should be highlighted in bold.
6. The format of reference 42 is written in capital letters, which does not comply with the formatting requirements of Nutrients for the reference list.
Author Response
We thank you very much for these further corrections, below there are the listed answers to your suggestions.
1. The correct abbreviation should be 'mL' instead of 'ml', and the authors should make this correction. For example, lines 20-21; line 30; lines 123-124; lines 159-161; lines 181; line 213.
Thank you very much for this suggestion. We updated the document as requested.
2. The abbreviations for “25-hydroxyvitamin D“ from lines73 to 81 are still inconsistent, and “3” for "D3" should be formatted as a subscript.
Thank you. We updated the document as requested.
3. Line 221, it should be“[33, 34]”not “[33], [34]”.
Thank you, we updated the document according to your suggestion.
4. All tables and figuresshould have consistent capitalization. CG: Control Group; CD: Celiac Disease; T1D: Type 1 Diabetes.
Thank you, we updated the document according to your suggestion.
5. The publication years of references 11, 13, 22 and 23 still need to be revised, which should be highlighted in bold.
Thank you for this useful suggestion. The data have been updated.
6. The format of reference 42 is written in capital letters, which does not comply with the formatting requirements ofNutrients for the reference list.
Thank you for this useful suggestion. The data have been updated.